# Genomic, Morphological and Biological Traits of the Viruses Infecting Major Fruit Trees

**DOI:** 10.3390/v11060515

**Published:** 2019-06-04

**Authors:** Muhammad Umer, Jiwen Liu, Huafeng You, Chuan Xu, Kaili Dong, Ni Luo, Linghong Kong, Xuepei Li, Ni Hong, Guoping Wang, Xudong Fan, Ioly Kotta-Loizou, Wenxing Xu

**Affiliations:** 1State Key Laboratory of Agricultural Microbiology, Wuhan 430070, China; muhammad.umer@webmail.hzau.edu.cn (M.U.); ljw13@webmail.hzau.edu.cn (J.L.); youhuafeng@webmail.hzau.edu.cn (H.Y.); Chuanxu@webmail.hzau.edu.cn (C.X.); Chuanxu@webmail.hzau.edu.cn (K.D.); konglinghong@webmail.hzau.edu.cn (L.K.); lixuepei@webmail.hzau.edu.cn (X.L.); whni@mail.hzau.edu.cn (N.H.); gpwang@mail.hzau.edu.cn (G.W.); 2Lab of Key Lab of Plant Pathology of Hubei Province, Wuhan 430070, China; 3College of Plant Science and Technology, Huazhong Agricultural University, Wuhan 430070, China; 4Key Laboratory of Horticultural Plant Biology (Ministry of Education), College of Horticulture and Forestry Sciences, Huazhong Agricultural University, Wuhan 430070, China; Niluo@webmail.hzau.edu.cn; 5Key Laboratory of Horticultural Crop (Fruit Trees) Biology and Germplasm Creation of the Ministry of Agriculture, Wuhan 430070, China; 6National Center for Eliminating Viruses from Deciduous Fruit Trees, Research Institute of Pomology, Chinese Academy of Agricultural Sciences, Xingcheng 125100, China; fxdong@163.com; 7Department of Life Sciences, Faculty of Natural Sciences, Imperial College London, London SW7 2AZ, UK

**Keywords:** fruit trees, virus, viral disease, taxonomy, viral genome, virion morphology, biological features

## Abstract

Banana trees, citrus fruit trees, pome fruit trees, grapevines, mango trees, and stone fruit trees are major fruit trees cultured worldwide and correspond to nearly 90% of the global production of woody fruit trees. In light of the above, the present manuscript summarizes the viruses that infect the major fruit trees, including their taxonomy and morphology, and highlights selected viruses that significantly affect fruit production, including their genomic and biological features. The results showed that a total of 163 viruses, belonging to 45 genera classified into 23 families have been reported to infect the major woody fruit trees. It is clear that there is higher accumulation of viruses in grapevine (80/163) compared to the other fruit trees (each corresponding to less than 35/163), while only one virus species has been reported infecting mango. Most of the viruses (over 70%) infecting woody fruit trees are positive-sense single-stranded RNA (+ssRNA), and the remainder belong to the -ssRNA, ssRNA-RT, dsRNA, ssDNA and dsDNA-RT groups (each corresponding to less than 8%). Most of the viruses are icosahedral or isometric (79/163), and their diameter ranges from 16 to 80 nm with the majority being 25–30 nm. Cross-infection has occurred in a high frequency among pome and stone fruit trees, whereas no or little cross-infection has occurred among banana, citrus and grapevine. The viruses infecting woody fruit trees are mostly transmitted by vegetative propagation, grafting, and root grafting in orchards and are usually vectored by mealybug, soft scale, aphids, mites or thrips. These viruses cause adverse effects in their fruit tree hosts, inducing a wide range of symptoms and significant damage, such as reduced yield, quality, vigor and longevity.

## 1. Introduction

Banana trees (banana and plantain), citrus fruit trees (orange, lemons, limes, grapefruit, and tangerine), pome fruit fruits (apple, quince, and pear), grapevines, mango trees, and stone fruit trees (peach/nectarine, apricot, plum, almond, and cherry) have a global production of 147.33, 108.01, 74.28, 129.76, 50.64 and 43.13 million metric tons, respectively, according to the statistical data published in 2017 (Statista Global Consumer Survey, provided by Statista GmbH, Hamburg, Germany). The total production of these fruit trees is 553.15 million metric tons and accounts for 86.55% worldwide, illustrating their major significance.

Citrus fruit trees (genus *Citrus*, family Rutaceae), including oranges, lemons, grapefruits, tangerines, and limes, are native to Southeast Asia, New Caledonia, and Australia, and are now grown throughout the tropics and subtropics worldwide [1]. Citrus fruit trees are mostly cultivated by clonal propagation, resulting in the accumulation of a large number of viruses which lead to severe losses of the susceptible rootstock and/or scion [2]. Banana trees and plantains (genus *Musa*, family Musaceae) are native to tropical Indomalaya and Australia [1], and nowadays are produced in over 130 countries, across the tropics in Africa, Asia, America, Oceania, and the Pacific, ranking among the world’s top 10 food crops. Banana and plantain trees are prone to the accumulation of viruses and are highly sensitive to virus infection, resulting in significant yield reductions and barriers to the international exchange of germplasm. Pome fruit (apple, quince, and pear) and stone fruit (apricot, peach, plum, almond, and cherry) trees are temperate and belong to the family Rosaceae. Their geographical distribution is highly influenced by environmental conditions since they require low temperatures to “break” dormancy and germinate. Apple trees, arguably, the most important fruit trees of the temperate world, originated in Central Asia, specifically Kazakstan; quinces originated in South-West Asia, Armenia, Turkey, Georgia, northern Iran to Afghanistan; and pears originated in western China, Asia Minor to the Middle East, and Central Asia [1]. Stone fruit trees originated in Central Asia, with secondary centers in Eastern Asia, Europe, and North America [1]. Nearly 50% of the total global volume of temperate tree fruits is produced in China, Turkey, USA, Brazil, Italy, and Spain. Temperate fruits contribute significantly to human nutrition as their daily consumption may reduce the risk of cardiovascular diseases and certain types of cancer [3]. Most pome and stone fruit trees do not root from cuttings; therefore, vegetative propagation is difficult without grafting, and almond is cultivated from seed. Pome and stone fruit trees are highly infected by viruses, resulting in significant economic losses to all sectors of the production chain [3]. Grapevines (genus *Vitis*, family Vitaceae) originated in the south Caspian belt, Turkey, and the Balkans [1], and are now mainly cultivated in Asia, North America, and Europe (tropical and subtropical areas). Grapevines are propagated vegetatively and are seriously affected by viral diseases. Mango (*Mangifera indica*, family Anarcardiaceae) is native to East Asia, specifically the Indo-Burma region [1], and is currently cultivated in most frost-free tropical and warmer subtropical areas, with almost 50% produced in India, followed by China, Thailand, Indonesia and Mexico. Mango is less affected by viral diseases and only one asymptomatic virus infection has been reported.

With the exception of almond, which is cultivated from seed, fruit trees have been traditionally multiplied using clonal propagation. This guarantees that the propagated plants have the same desirable traits as the parent; for instance, reduced plant juvenility phase in citrus so that fruits are produced sooner. Because of this clonal propagation, fruit trees have accumulated a large number of viruses that may be latent or may cause detectable symptoms in susceptible rootstocks and/or scions. These viruses belong to numerous different genera and families, have isometric or filamentous virions and their genomes may be DNA or RNA, double-stranded (ds) or single-stranded (ss), positive-sense (+) or negative-sense (–), linear or circular. The severity of the disease and the symptoms caused, and subsequently the agricultural impact on the fruit production, may also vary. The viruses occurring in major fruit trees, together with their taxonomy and molecular and biological traits, are reviewed here.

## 2. Viruses Infecting Fruit Trees

### 2.1. Viruses Infecting Citrus

Twenty-two viruses, belonging to fifteen genera classified into fifteen families, have been identified infecting citrus (Appendix A).

Citrus tristeza virus (CTV; genus *Closterovirus*, family *Closteroviridae*) is a flexuous filamentous virus (2000 × 11^−12^ nm in size), with a +ssRNA genome (approximately 20 kb in length), containing twelve open reading frames (ORFs) flanked by 5′- and 3′-untranslated regions (UTRs) [4]. CTV is probably the most economically important virus infecting citrus, causing decline of sour orange rootstock, yellow seedling of lemon and grapefruit, and stem pitting in grapefruit and sweet orange [2]. CTV is transmitted in a semi-persistent manner by a number of aphid species (for instance, *Aphis gossypii* and *Toxoptera citricida*) [2].

Citrus psorosis ophiovirus (CPV; genus *Ophiovirus*; family *Aspiviridae*) is a -ssRNA virus consisting of three genomic segments (8186, 1645, and 1447 nt in length, respectively) encapsidated in filamentous nucleocapsids (approximately 760 × 3–4 nm in size) [5]. CPV causes bark scaling lesions on the trunk and branches of sweet orange, mandarin, and grapefruit after the infected citrus trees are grown for 3–7 years or longer [5].

Citrus leprosis virus C, C2 and N (CiLV-C and -C2; +ssRNA genome, genus *Cilevirus*; CiLV-N, -ssRNA genome, genus *Dichorhavirus*, family *Rhabdoviridae*) have vastly different bipartitie genomes (8729/4969, 8717/4989 and 6268/5847 nt in length for CiLV-C, -C2 and -N, respectively) and membrane-bound bacilliform virions (120–130 × 50–55 nm in size in case of CiLV-C) [6,7]. They cause similar symptoms in the citrus hosts and are transmitted by the same vector, mites of the genus *Brevipalpus* in a persistent manner [8]. All three are non-systemic; CiLV-C is limited to the cytoplasm, specifically the endoplasmic reticulum, while CiLV-N is found in the nuclei and cytoplasm of infected cells [2]. CiLVs have significant economic impacts on citrus crops including oranges, grapefruits, and tangerines, resulting in stunted growth and one-third or more losses of fruit yields. CiLVs cause similar symptoms in the citrus hosts, including white to yellow-green flecks, spots, rings, or large translucent areas on young leaves, and rings bordered by sunken grooves on the fruits. In host trees older than six years, CiLVs cause scaly bark, or small irregular pustules and gum like deposits on the outer bark, internal staining in the underlying wood, and variously sized cavities or narrow grooves in the large branches and trunk [9]. All three viruses are transmitted by the same vector, mites of the genus *Brevipalpus*, in a persistent manner [8].

Citrus vein enation virus (CVEV; genus *Enamovirus*, family *Luteoviridae*) is a +ssRNA virus (5983 nt in length) with spherical particles (28 nm in diameter) [10]. CVEV is associated with enations in the leaf veins of sour orange, and woody galls on the trunks or rootstocks of acid lime, rough lemon, Rangpur lime, and Volkamer lemon [10]. The disease is transmitted in a persistent manner by several aphid species including *T. citricida*, *Myzus persicae*, and *A. gossypii* [8,10].

Citrus tatter leaf virus (CiTLV; a.k.a. apple stem grooving virus; genus *Capillovirus*, family *Betaflexiviridae*) is a flexuous filamentous virus (600–700 × 12 nm in size), with a +ssRNA genome (6496 nt in length excluding the poly (A) tail). CiTLV causes stunting or dwarfing, necrosis at the bud union, and virus-induced bud union incompatibility on scions grafted onto *P. trifoliata*, citrange, or citrumelo rootstocks [11]. The symptoms often become apparent after the trees are cultured for 3–7 years.

Citrus leaf blotch virus (CLBV; genus *Citrivirus*, family *Betaflexiviridae*) has filamentous virions (approximately 960 × 12–15 nm in size) and a +ssRNA genome (8747 nt in length). CLBV was associated with bud union crease of sweet orange, grapefruit and Clementine plants grafted on *P. trifoliata* or trifoliate hybrid rootstocks [2].

Citrus yellow vein clearing virus (CYVCV, genus *Mandarivirus*, family *Alphaflexiviridae*) is a filamentous virus (960 × 14 nm in size) with a +ssRNA genome (7531 nt in length). CYVCV occasionally causes yellow veins, ringspots and venial necrosis in the leaves, irregular yellow blotches in the rind, and smaller and flatter fruits on many varieties, especially on limes and lemons. CYVCV is considered an emerging threat for the citrus fruit industry [6,12,13].

Citrus yellow mosaic virus (CYMV; genus *Badnavirus*, family *Caulimoviridae*) is a nonenveloped bacilliform virus (150 × 30 nm in size) with a dsDNA genome (7559 bp in length) [14]. CYMV infects most citrus cultivars and their related plants, causing strong symptoms on oranges, grapefruit, and mandarins, but not on Mexican lime. The symptoms include bright yellow mottling for vein flecking on mature leaves, and occasionally on fruits. CYMV is spread by infected propagating source materials and the citrus mealybug, *Planococcus citri* [14].

Satsuma dwarf virus (SDV; genus *Sadwavirus*, family *Secoviridae*) has an isometric virion (26 nm in diameter) and two +ssRNAs (6790 and 5345 nt, respectively) as its genome [15]. SDV causes spoon-shaped leaves, enations, multiple flushing, stunting or dwarfing, fewer leaves, and small fruit having a thick peel [16]. Interestingly, SDV has one variant termed citrus mosaic virus found exclusively in Japan that induces leaf mosaic symptoms. SDV is transmitted by grafting, and no vector has been reported.

Citrus variegation virus (CVV; genus *Ilarvirus*, family *Bromoviridae*) has a tripartite +ssRNA genome (3433, 2914 and 2309 nt in length, respectively). CVV causes a range of symptoms; usually mild on oranges and mandarins but potentially severe on citrons and lemons, associated with fruit malformation and yield reduction [17]. Interestingly, two CVV strains cause infectious variegation and crinkly leaf, respectively. More specifically, the former symptoms include crinkling of leaves associated with areas displaying various degrees of chlorosis, while leaves may be narrower and have an irregular outline. The latter causes warping, pocketing and crinkling without variegation or reduction in leaf size [17].

Citrus endogenous pararetrovirus (CitPRV; family *Retroviridae*) and citrus sudden death-associated virus (CSDaV; genus *Marafivirus,* family *Tymoviridae*) may be associated with citrus sudden death disease, affecting four million orange trees in a very important citrus region in Brazil [18]. Citrus jingmen-like virus (CJLV) and citrus virga-like virus (CVLV) are two novel viruses recently identified, which appear to have no obvious impact on citrus plants [18]. In 2018, the first phlebo-like virus infecting plants was identified in citrus; citrus concave gum-associated virus (CCGaV) may represent a new genus. CCGaV is flexuous and non-enveloped (200–300 × 6 nm in size), and probably contains two RNA components (6681 and 2703 nt in length, respectively). CCGaV is closely associated with a severe citrus disease, concave gum-blind pocket, in sweet orange, mandarin and clementine [19]. The remaining viruses are of minor significance or to date there is not enough research data available to access their significance.

### 2.2. Viruses Infecting Pome Fruits

Twenty-one viruses, belonging to twelve genera classified into nine families, have been identified infecting pome fruit trees (Appendix A).

Apple chlorotic leaf spot virus (ACLSV, genus *Trichovirus*), apple stem grooving virus (ASGV; genus *Capillovirus*) and apple stem pitting virus (ASPV; genus *Foveavirus*) all belong to the family *Betaflexiviridae*, and have +ssRNA genomes (7555, 6495 and 9332 nt in length, respectively) encapsidated in filamentous virions (640–890 × 10–12 nm, 620–680 × 12, and 800 × 12–15 nm in size, respectively). These viruses have similar molecular and biological features, and while they remain generally latent in most commercial apple cultivars, they may cause disease when grafted on sensitive rootstocks [20]. ACLSV infects most fruit trees of Rosaceae family, which include apple, pear, quince, sweet and sour cherry, peach, plum and apricot; it reduces tree vigor (50% on pear), yield (40% on pear), and quality of fruits. ASGV also reduces vigor and causes stem grooving, brown lines, and graft union abnormalities when an infected cultivar is grafted on sensitive rootstocks [21]. ASPV causes a variety of symptoms including xylem pitting, topworking disease, epinasty and lethal decline on susceptible apple cultivars or when grafted on sensitive rootstocks [3]. All three viruses are transmitted by infected propagative material and grafting, and it is not known whether they are transmitted by seed, pollen or natural vectors [20].

Apple mosaic virus (ApMV; genus *Ilarvirus*, family *Bromoviridae*) has a tripartite +ssRNA genome (3476, 2979 and 2056 nt in length, respectively) encapsidated in icosahedral virions (26–35 nm in diameter). ApMV infects a large number of woody hosts and is frequently found in mixed infections with ACLSV, ASPV, ASGV, and other apple-infecting viruses. ApMV causes pale yellow to bright cream irregular spots or bands together with major veins on spring apple leaves, with the severity depending on cultivar susceptibility. The symptomatic leaves may be distributed randomly on the tree or limited to a single branch, and drop prematurely. ApMV is asymptomatic in pear [22]. ApMV is transmitted by vegetative propagation and by grafting, and no insect vector is known [22,23,24].

A monopartite circular ssDNA virus (3442 nt in length) named temperate fruit decay-associated virus (TFDaV; unassigned genus and family) was identified infecting apple, pear and grapevine trees in Brazil [25]. TFDaV causes no foliar symptoms but causes growth reduction of infected pear and apple (cvs. Gala and Fuji) [25].

The remaining viruses infecting pome fruit trees are of minor significance or to date there is not enough research data available to access their significance.

### 2.3. Viruses Infecting Stone Fruits

Thirty-five viruses, belonging to fifteen genera classified into nine families, have been identified infecting stone fruit trees (Appendix A).

Plum pox virus (PPV; genus *Potyvirus*, family *Potyviridae*) has a +ssRNA genome (approximately 9786 nt in length) encapsidated in filamentous virions (750 × 15 nm in size), and causes ‘’sharka’’ disease, the most devastating disease of stone fruit trees worldwide which causes severe damages and has enormous economic and social impact. Tens of millions of euros and dollars have been spent for controlling this pathogen without success [26]. PPV is responsible for the appearance of pale or yellowish green rings, spots, or mottling on the leaves of the susceptible stone fruit trees, including peach, cherry, apricot, and plum cultivars [27]. Moreover, PPV also causes rings, irregular lines, and poxes on the plum fruit surface, and deformation of plum fruits; coloured rings and bands on the skin of apricot fruits and pale rings or spots on apricot stones; pale rings and diffuse bands on the peach fruit skins before maturation [27]. Nine PPV strains or types have been identified depending on their biological, serological, and molecular characteristics [28,29]. PPV is transmitted by vegetative propagation and grafting, and by aphids (e.g., *A. spiraecola* and *M. persicae*) in a nonpersistent manner. There is no evidence for either pollen or seed transmission [30].

ApMV (described in the section ‘Viruses infecting pome fruits’), prunus necrotic ringspot virus (PNRSV), and prune dwarf virus (PDV) all belong to the genus *Ilarvirus*, family *Bromoviridae*, contain tripartite genomes (3332, 2591 and 1957 nt in length for PNRSV and 3374, 2593, and 2129 nt in length for PDV) and are frequently found in mixed infections on stone fruit hosts. PNRSV is a nonenveloped isometric and quasi-isometric (bacilliform) virus (23, 25, and 27 nm in diameter), while PDV is a multicomponent virus with five types of particles differing in size (quasi-isometric about 19–20 nm in diameter, and bacilliform up to 73 nm in length) [31]. Both viruses are relatively unstable in tissue extracts, a feature of ilariviruses. The three ilariviruses cause economic losses on stone fruit trees, especially in sour and sweet cherry, almond, and peach [32,33]. For instance, both PDV and PNRSV infection cause bark splitting, and can reduce yield by up to 60% for peach cultivars. The severity of the symptoms depends on the specific virus isolates, host cultivars, and their synergistic interactions ApMV causes a typical yellow line pattern, bright yellow blotches, rings, bright yellow vein clearing, and/or oak-leaf pattern in stone fruits [34,35]. PDV causes mild stunting, dark green and more erect leaves, and reduced yield and quality of peach fruits in some cultivars; stunting and leaf malformation and shortened internodes in plum; leaf chlorotic spots, rings and diffuse mottling, possibly stem pitting and flat limb, malformed fruits, and reduced production in cherry; and gummosis on the trunk in some apricot cultivars. PNRSV is easily transmitted by vegetative propagation and grafting, root grafting in orchards, pollen grains and seeds, and different thrip species; ApMV is transmitted by vegetative propagation grafting, and root grafting; and PDV is transmitted by grafting (buds, scions), pollen and seed [3].

Cherry green ring mottle virus (CGRMV; genus *Robigovirus*, family *Betaflexiviridae*) has a +ssRNA genome (8372 nt in length) encapsidated in filamentous virions (1000–2000 nm in length). CGRMV is generally latent in most *Prunus* species, and causes green islands and rings cast against a yellow mottle background on leaves, together with misshapen and bitter fruits with pitted and necrotic fruit flesh [36]. CGRMV is transmitted by vegetative propagation and root grafting [3].

ACLSV (described in the section ‘Viruses infecting pome fruits) is associated with severe fruit deformations, yield reduction, graft incompatibility, and bud necrosis in some stone fruit species. Some ACLSV isolates cause “pseudopox” disease in apricot and plum, characterized by depressions and protuberances that deform the fruit, which is easily confused with the PPV-caused “sharka” disease [3].

Cherry mottle leaf virus (CMLV; genus *Trichovirus*, family *Betaflexiviridae*) has a +ssRNA genome (8018 nt in length) encapsidated in filamentous virions (640–800 × 12 nm in size, with striations 3 nm in pitch), and represents the most severe disease on cherry cultivars in some regions. CMLV causes chlorotic mottling and distortion of the leaves, and smaller, flavorless and delayed ripening fruits. CMLV can be transmitted by grafting and also efficiently by the bud/scale mite *Eriophyes inaequalis* [34].

Tomato ringspot virus (ToRSV; genus *Nepovirus*, family *Secoviridae*) has a bipartite +ssRNA genome (8214 and 7271 nt in length, respectively) encapsidated in icosahedral virions (28 nm in diameter). ToRSV is considered one of the most devastating members of the genus with a worldwide distribution. The virus is found in woody and semi-woody hosts, and also in herbaceous ornamental and weedy species [3]. Major diseases caused by ToRSV on fruit crops include yellow bud mosaic in peach and almond, whose symptoms are pale-green to pale-yellow blotches along the main vein or large lateral veins of leaves [3]. Plants infected with ToRSV show severe distinctive symptoms as a shock reaction, and some strains of the virus cause stem pitting and decline in *Prunus* spp [24]. In addition to peach and other stone fruit trees, strains of ToRSV have been isolated from apple, grape, tobacco, and euonymus [3]. ToRSV may cause necrosis of the union in apple, grapevine decline, which is economically important in New York, USA and yellow stripe of lilies [33].

Little cherry virus 1 and 2 (LChV-1 and LChV-2; genera *Velarivirus* and *Ampelovirus*, respectively, family *Closteroviridae*) have +ssRNA genomes (16,930 and 15,050 nt in length, respectively) encapsidated in filamentous virions (1786–1820 nm in length and 1667 × 11 nm in size, respectively). Both viruses are associated with little cherry disease, which is a complex and serious viral disease with a large impact on fruit quality of infected cherry trees worldwide. LChV-1 and LChV-2 can dramatically reduce size, color, and taste of the fruits, and a damage vegetative growth of susceptive cherry varieties especially as mixed infections [37]. Both viruses are transmitted by grafting, While LChV-2 is also transmitted by mealybugs (*Phenacoccus aceris* and *Pseudococcus maritimus*) [38].

Recently, thirteen novel viruses belonging to eight families have been reported by next-generation sequencing from stone fruit trees [39,40], and contribute to better understanding of viruses infecting stone fruit trees. For instance, peach leaf pitting-associated virus (PLPaV) was the first reported fabavirus infecting peach and is associated with leaf pitting symptoms and also demonstrates several novel molecular and biological features that are absent in other fabaviruses [39]. Most of these novel viruses impact on the stone fruits requires further assessment.

### 2.4. Viruses Infecting Grapevine

A total of 80 viruses (including five unnamed ones), belonging to twenty-nine genera classified into eighteen families, have been identified infecting grapevines. Some of them cause serious diseases including leafroll, rugose wood (RW), fanleaf, decline and other symptoms [41] (Appendix A).

Grapevine leafroll-associated viruses (GLRaVs) 1, 2, 3, 4, 7 and 13 are filamentous viruses belonging to the family *Closteroviridae* (GLRaVs 1, 3 4 and 13; genus *Ampelovirus*; GLRaV-7; genus *Velarivirus*; GLRaV-2; genus *Closterovirus*), with +ssRNA genomic segments (18,659, 16,494, 17,919, 13,830, 6404, and 17,608 nt in length, respectively) encapsidated in long flexuous virions (1350–2000 × 12 nm in size). GLRaVs are associated with grapevine leafroll disease (GLRD), which is characterized by reddening (red-berried cultivars) or yellowing (white-berried cultivars) of the leaves, while the veins remain green and the leaf margins roll downward in late summer and autumn, resulting in decreased vigour and yield by 15–20% on average [42]. GLRaV-1, -2 and -3 cause strong leafroll symptoms, whereas GLRaV-4 is milder and GLRaV-7 causes no or very mild symptoms [43]. Some GLRaV-2 strains (for instance GLRaV-2 RG) are also involved in graft incompatibility on certain rootstocks [44]. GLRaVs 1, 3 4 and 13, belonging to the genus *Ampelovirus*, are transmitted through vegetative propagation, grafting, and insect vectors including mealybugs and soft scale species, while no vector has been identified for GLRaV-7 (genus *Velarivirus*), GLRaV-2 (genus *Closterovirus*), and GLRaV-4 strain Car [44].

Grapevine virus A, -B, and -D (GVA, GVB, and GVD; genus *Vitivirus*), and grapevine rupestris stem pitting-associated virus (GRSPaV; genus *Foveavirus*) belong to the family *Betaflexiviridae*. They have +ssRNA genomes (7351, 7599, and 7479 nt in length, respectively) encapsidated in filamentous virions (600–1000 × 10–15 nm in size). GVA, GVB, and GVD are associated with the RW complex symptoms, characterized by alterations of the woody cylinder such as pits and grooves on the scion or/and rootstock, together with an overall decrease in vigour and yield, and possibly graft incompatibility [45]. Similar to the GLRaVs, GVA and GVB are transmitted through vegetative propagation and grafting, and by mealybugs and soft scale in semi-persistent manner, whereas GVD and GRSPaV have no known vector [46].

Grapevine fanleaf virus (GFLV; genus *Nepovirus*, family *Secoviridae*) has a bipartite +ssRNA genome (3774 and 3342 nt in length, respectively) encapsidated in icosahedral virions (30 nm in diameter). GFLV is the main etiological agent of fanleaf degeneration disease, characterized by a distortion with toothed margins, closer primary veins, and widely open petiolar sinuses [40,47]. GFLV causes two distinct symptoms, malformations and bright yellow discolorations, depending on the strain [42]. Additionally, GFLV causes short internodes, double nodes, and zig–zag growth between nodes of canes, and significantly decreases the fruit quality and the productive life, resulting in severe losses up to 80% [48,49]. GFLV is semi-persistently transmitted by a dagger nematode (*Xiphinema index*) that feeds on roots [49].

### 2.5. Viruses Infecting Mango

The only virus reported infecting mango, mangifera indica latent virus (MILV), is not associated with pathological symptoms on mango fruits or leaves [50]. This novel bipartite virus was discovered by sequencing the mango transcriptome. MILV is a +ssRNA virus belonging to genus *Benyvirus* of the family *Benyviridae* and is phylogenetically related to beet soil-borne mosaic virus, beet soil-borne mosaic virus, rice stripe necrosis virus, and burdock mottle virus, since the MILV RNA1 coding protein shares a sequence homology of 38–39% with those of these benyviruses [50].

### 2.6. Viruses Infecting Banana

Eighteen viruses, belonging to four genera classified into six families, have been identified infecting banana worldwide (Appendix A). These are a major concern to producers due to their significant effects on yield and quality.

Banana bunchy top virus (BBTV; genus *Babuvirus*, family *Nanoviridae*) is a circular ssDNA virus (18–20 nm in diameter), consisting of six different components referred to as DNA-R, -S, -C, -M, -N, and -U3 (each 1060 ± 50 nt in length). BBTV is the most devastating virus; it causes banana bunchy top disease and leads to severe reductions in the productivity. For instance, a yield loss of 50–100% in severely infected plants of highly susceptible cultivars was reported from affected regions [51]. The characteristic symptoms are discontinuous dark green flecks and streaks of variable length on the leaf sheath, midrib, leaf veins, and petioles; new leaves are narrower with wavy leaf lamina and yellow leaf margins, shorter and brittle in texture; suckers are severely stunted [52]. Severely infected banana plants usually produce no fruits, or produce fruits with distorted and twisted hands and fingers [52]. BBTV is transmitted through vegetative propagules, and by banana aphids (*P. nigronervosa* and *P. caladii*) in a persistent and circulative manner [51].

Banana streak viruses (BSVs; genus *Badnavirus*, family *Caulimoviridae*), including banana streak GF virus, banana streak IM virus, banana streak MY virus, banana streak OL virus, banana streak UA virus, banana streak UI virus, banana streak UL virus, banana streak UM virus, banana streak VN virus) have bacilliform-shaped virions (130–300 × 30 nm in size), and noncovalently closed circular dsDNA genomes (approximately 7.2–7.8 kb in length). BSVs are responsible for banana streak disease, which is widely distributed worldwide. BSVs infect the natural and synthetic hybrids of *Musa* plants and cause initially yellow and finally necrotic dots or streaks on the leaves, and pseudo-stem splitting. Some isolates may cause stunting, leaf necrosis and a premature death [53]. BSVs are transmitted by the use of infected planting material and are vectored by mealybugs *(Planococcus* spp., *Pseudococcus* spp., *Dysmicoccus* spp., *Ferrisia virgata* and *Paracoccus burnerae*) in a semipersistent manner [54,55].

Banana bract mosaic virus (BBrMV; genus *Potyvirus*, family *Potyviridae*) has a +ssRNA genome (9711 nt in length excluding the 3ʹ-terminal poly(A) tail) encapsidated in flexuous and filamentous particles (660–760 × 12 nm in size). BBrMV is among the most economically important viruses, resulting in yield losses ranging from 30% to 70% for infected banana plants according to previous records [45,47]. BBrMV causes spindle-shaped and purplish streaks on bracts, pseudostems, midribs, peduncles, and even fruits that subsequently turn into necrotic tissues on the fruits, leaves, pseudostems, and midribs [56]. BBrMV is primarily transmitted through infected plant material and is vectored by aphids (*P. nigronervosa*, *Rhopalosiphum maidis*, *A. gossypii*, and *A. craccivora*) in a nonpersistent manner [56].

Abaca bunchy top virus (ABTV; genus *Babuvirus*, family *Nanoviridae*), sugarcane mosaic virus (SCMV) strain SCMV-Ab (genus *Potyvirus*, family *Potyviridae*), banana mild mosaic virus and banana virus X (BanMMV and BVX; family *Betaflexiviridae*), and cucumber mosaic virus (CMV, genus *Cucumovirus*, family *Bromoviridae*) are of minor significance [51].

## 3. Summary

In total, there are 163 viruses, belonging to forty-five genera classified into twenty-three families, which have been reported to infect the woody fruit trees including banana, citrus, pome fruits, grapevine, mango, and stone fruits (Appendix A). It is clear that viruses accumulate much more in grapevine (80/163) than in other fruit trees (less than 35/163), which might be closely related to highly vegetative propagation of grapevine. Correspondingly, only one virus species has been reported infecting mango. The limited viruses infecting mango might be due to it being cultured in tropical climates with high temperature, which may increase the host’s antiviral defenses and interfere with virus replication [57]. Alternatively, mango seedlings are generally multiplied by grafting variety buds onto the seed-developed seedlings, a process which may also prevent virus accumulation. The numbers of viruses infecting citrus, banana, pome and stone fruit trees are similar, ranging from eighteen to twenty-three. Five viruses, ApMV, ACLSV, ApLV, CLRV and ToRSV, infect both pome and stone fruit trees, whereas three to four viruses co-infect grapevine and pome or stone fruit trees. One virus, CCGaV, infects both citrus and pome fruit trees. In contrast, no viruses have been found to co-infect banana and the other fruit trees, with the exception of CMV, a virus with extremely wide host range that infects both banana and grapevine (Figure 1). This observation suggests that cross-infections have occurred in high frequency among pome and stone fruit trees, whereas no or little cross-infection has occurred among banana, citrus and grapevine. This might be due to that the pome and stone fruit trees belong to the same family and share the same insect vectors, facilitating the cross-infections.

The majority of the viruses, over 70%, infecting woody fruit trees have +ssRNA genomes, while the remaining viruses belong to the -ssRNA, ssRNA-RT, dsRNA, ssDNA and dsDNA-RT (Appendix A). Approximately half of the viruses are icosahedral, and their diameter ranges from 16 to 120 nm with most being 25–30 nm (Appendix A). The remaining viruses are mainly filamentous, ranging from 200 to 2000 nm in length and 3 to 16 nm in width, or bacilliform, with lengths of 120–150 and widths of 30–55 nm. Most of the viruses contain one genomic component, followed by two and three genomic components. The length of monopartite genomes ranges from 2.5 to 20 kb(p), while bipartite are 3–15 kb(p), and tripartite 7.5–17 kb(p). Only two viruses have six and ten genomic components, respectively, with genomes approximately 6–6.5 kb and 25 kbp. For some of the viruses, especially those discovered by next-generation sequencing, full information on the number and length of their genomic segments may not be available.

The major woody fruit trees, except mango, are extensively infected by different virus species, with some resulting in important economic losses by reducing the yield and quality of the fruits, decreasing crop vigour, increasing of the sensitivity to frost and drought, and shortening the life span. These viruses can cause obvious symptoms on many parts of the susceptible fruit trees, including leaves and fruits. The leaf symptoms include yellows, chlorosis, mottling, vein enations, yellow veins, necrotic dots or streaks, and leaf and bud necrosis; white to yellow-green flecks, spots, rings, or large translucent areas in young leaves; warping, pocketing and crinkling without variegation or reduction in leaf size, distortion of the leaves. The fruit symptoms include appearance of rings, irregular lines, poxes or spots on the fruit surface together with dramatic alterations in size, colour, and taste of the fruits. Additionally, the symptoms on the trunk and branches include stem grooving, stem pitting, bark scaling lesions, scaly bark, graft union abnormalities, shortened internodes, gummosis, bark splitting, and woody galls. Finally, the systematic symptoms include stunting or dwarfing, multiple flushing, early defoliation, graft incompatibility, reduction in vigor, leaves, vegetative growth, fruit yield and quality, and rapid death of the plant. The viruses infecting woody fruit trees are mostly transmitted by vegetative propagation, grafting (buds, scions), and root grafting in orchards. Some of them are also transmitted by pollen grains and seeds and are vectored by mealybugs, soft scales, aphids, mites or thrips.

The review summarizes the currently available information on the biological, genomic and morphological traits of viruses infecting woody fruit trees, which can be utilized to help the prevention and control of numerous ecologically and economically important diseases. Detailed knowledge of the viral diseases potentially affecting specific woody fruit trees will lead to faster identification of the causative agents. Additionally, known genomic sequences will facilitate the design of oligonucleotide primers targeting specific viruses or groups of viruses enhancing detection.

## Figures and Tables

**Figure 1 viruses-11-00515-f001:**
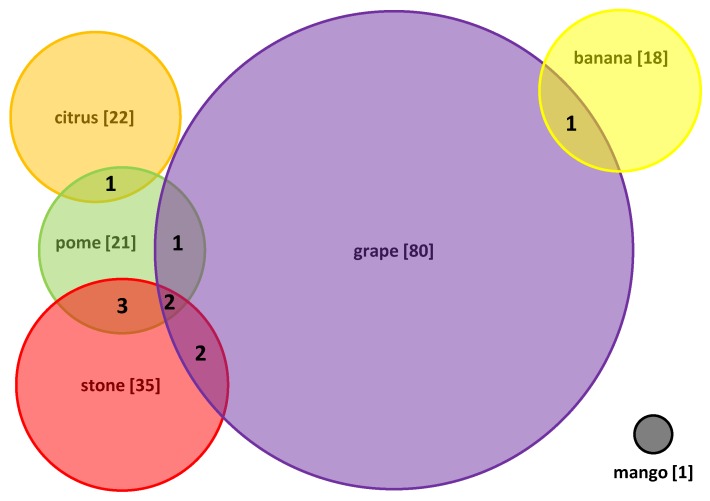
Venn diagram illustrating virus cross-infections among woody fruit trees.

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
