# Peer review of "Genomic, Morphological and Biological Traits of the Viruses Infecting Major Fruit Trees"

_viruses, 2019, doi:10.3390/v11060515_

Reviewer 1 Report

The paper of Muhammad and co-authors compiles findings about viruses on major fruit trees. Indeed, the number of identified viruses is constantly increasing, and comprehensive up-to-date review will be of a great value for the field.

However, authors should carefully re-evaluate the selection of included viruses. They omitted some viruses on purpose due to 'minor significance or to date there is not enough research data available to access their significance', but it would be helpful to include other novel and/or not so well studied viruses at least into tables without detailed text description (although there are several such records – for instance, Table 4 'Two unnamed viruses'). That will impact all counts and percentages throughout the text.

The title and abstract should not contain 'molecular', as the molecular characterization is rather limited and covers kinds and sizes of genome.

The manuscript will be interesting for the field and after taking into account above-mentioned points could be published as a review.

General comments:

- check virion sizes, there are several sizes less than 10 nm (for instance, lines 105, 179, and Table 3)

- viral genera and families names should be written with definite article

 line 114           the reference [2] about the presence of CiLV-N in cytoplasm and nucleus is incorrect and should be updated.

lines 372-378  consider a Venn diagram for representation of cross-infections

 line 389 '852 nt' hardly corresponds with a viral genome.

lines 408-410  it is not clear how exactly the review will help to prevent/control. The gathered data are rather compilation and to prove their practical usefullness, the authors should include information about diagnostic/detection methods.

References contains a lot of words written in lower case (RNA, Prunus, Ampelovirus, ...).

line 433           Virusdisease -> Virus' 'disease

 Tables:

- consider adding reference column;

- Virion/Structure shoul be rather Virion/'Morphology'

Author Response

The paper of Muhammad and co-authors compiles findings about viruses on major fruit trees. Indeed, the number of identified viruses is constantly increasing, and comprehensive up-to-date review will be of a great value for the field.

We thank the reviewer for their kind comments and for their insightful suggestions that helped us improve the manuscript.

 However, authors should carefully re-evaluate the selection of included viruses. They omitted some viruses on purpose due to 'minor significance or to date there is not enough research data available to access their significance', but it would be helpful to include other novel and/or not so well studied viruses at least into tables without detailed text description (although there are several such records – for instance, Table 4 'Two unnamed viruses'). That will impact all counts and percentages throughout the text.

Actually, as the reviewer already noted, many more viruses are included in the tables without detailed description in the text. For instance, regarding viruses infecting grapevine trees, we have listed 79 viruses in Table 4 and we only discuss 10 of them in the text. All the counts and percentages reported in the discussion were based on the overall data.

 The title and abstract should not contain 'molecular', as the molecular characterization is rather limited and covers kinds and sizes of genome.

Fair point. We have removed the word ‘molecular’ from the title of the manuscript and replaced with ‘genomic and morphological’. Additionally, we have modified the abstract as accordingly.

 The manuscript will be interesting for the field and after taking into account above-mentioned points could be published as a review.

 General comments:

- check virion sizes, there are several sizes less than 10 nm (for instance, lines 105, 179, and Table 3)

These sizes are correct based on the referenced literature. For instance, both citrus psorosis ophiovirus (CPV; line 105) and citrus concave gum-associated virus (CCGaV; line 179) have filamentous virions with one dimension less than 10 nm, while their other dimension is over 100 nm.

 - viral genera and families names should be written with definite article

We are a bit confused by this suggestion since we do not refer to genera and families using the indefinite article in the text. Could the reviewer further clarify what needs to be amended?

 line 114 the reference [2] about the presence of CiLV-N in cytoplasm and nucleus is incorrect and should be updated.

We thank the reviewer for pointing this out. We now reference Ramos-González et al., 2017.

 lines 372-378 consider a Venn diagram for representation of cross-infections

This is an excellent idea and we have now added a Venn diagram (Figure 1) in the revised manuscript.

 line 389 '852 nt' hardly corresponds with a viral genome.

We thank the reviewer for pointing this out. We have gone through our data again in detail and corrected any errors regarding genome size. It should be noted that for some of the viruses, especially those discovered by next-generation sequencing, full information on the number and length of their genomic segments may not be available.

 lines 408-410 it is not clear how exactly the review will help to prevent/control. The gathered data are rather compilation and to prove their practical usefulness, the authors should include information about diagnostic/detection methods.

Detailed knowledge of the viral diseases potentially affecting specific woody fruit trees will lead to faster identification of the causative agents. Additionally, known genomic sequences will facilitate the design of oligonucleotide primers targeting specific viruses or groups of viruses enhancing detection. PCR together with the use of antibodies against viruses are the most widely used diagnostic/detection methods and this information has now been added to the revised manuscript.

 References contains a lot of words written in lower case (RNA, Prunus, Ampelovirus, ...).

We thank the reviewer for pointing this out. We have now gone through the references and amended as appropriate.

 line 433 Virusdisease -> Virus' 'disease

Actually, this is not a typographical error: ‘Virusdisease’ is the abbreviated name of the journal.

 Tables:

- consider adding reference column;

- Virion/Structure should be rather Virion/'Morphology'

These are good suggestions and in the revised manuscript we have included a reference for the sequence of the virus, if available, and replace Virion/Structure with Virion/Morphology. We have also added a ‘Host’ column in the last table. Due to their inconvenient size, the tables are now designated as supplementary.

Reviewer 2 Report

The paper "Molecular and biological traits of the viruses infecting major fruit trees" is a review paper describing viruses infecting citruses, pome fruits, stone fruits, grapevines, mango and banana. It is a nice review, gathering information regarding virus species in mentioned fruits. I was not able to find tables, there should be six of them, therfore I didn't review them. Paper is well written, I recommend acceptance in the present form.

Author Response

We would like to thank this reviewer for taking the time to read our manuscript and we are grateful for their kind comments.

Round  2

Reviewer 1 Report

Nearly all suggestions were addressed. However, the tables still miss a number of recently reported viruses. For example:

Caglayan, K., Roumi, V., Gazel, M., Elci, E., Acioğlu, M., Mavrič Pleško, I., et al. (2019). Identification and Characterization of a Novel Robigovirus Species from Sweet Cherry in Turkey. Pathogens (Basel, Switzerland), 8(2). http://doi.org/10.3390/pathogens8020057

Cao, M., Li, P., Zhang, S., Yang, F., Zhou, Y., Wang, X., et al. (2018). Molecular characterization of a novel citrivirus from citrus using next-generation sequencing. Archives of Virology, 163(12), 3479–3482. http://doi.org/10.1007/s00705-018-4039-8

Marais, A., Faure, C., Theil, S., & Candresse, T. (2018). Molecular characterization of a novel species of capillovirus from Japanese apricot (Prunus mume). Viruses, 10(4). http://doi.org/10.3390/v10040144

Milusheva, S., Phelan, J., Piperkova, N., Nikolova, V., Gozmanova, M., & James, D. (2018). Molecular analysis of the complete genome of an unusual virus detected in sweet cherry (Prunus avium) in Bulgaria. European Journal of Plant Pathology, 153(1), 197–207. http://doi.org/10.1007/s10658-018-1555-z

Koloniuk, I., Sarkisova, T., Petrzik, K., Lenz, O., Přibylová, J., Fránová, J., et al. (2018). Variability Studies of Two Prunus-Infecting Fabaviruses with the Aid of High-Throughput Sequencing. Viruses, 10(4), 204–13. http://doi.org/10.3390/v10040204

Rwahnih, Al, M., Alabi, O. J., Westrick, N. M., & Golino, D. (2018). Prunus geminivirus A: a novel Grablovirus infecting Prunus spp. Plant Disease, 102(7), 1246–1253. http://doi.org/10.1094/PDIS-09-17-1486-RE

Lenz, O., Přibylová, J., Fránová, J., Koloniuk, I., & Špak, J. (2017). Identification and characterization of a new member of the genus Luteovirus from cherry. Archives of Virology, 162(2), 587–590. http://doi.org/10.1007/s00705-016-3125-z

Villamor, D. E. V., Pillai, S. S., & Eastwell, K. C. (2017). High throughput sequencing reveals a novel fabavirus infecting sweet cherry. Archives of Virology, 162(3), 811–816. http://doi.org/10.1007/s00705-016-3141-z

Wu, L. P., Liu, H. W., Bateman, M., Liu, Z., & Li, R. (2017). Molecular characterization of a novel luteovirus from peach identified by high-throughput sequencing. Archives of Virology, 91(9), 1–3. http://doi.org/10.1007/s00705-017-3388-z

Minor comments:

Abbreviations of viral species should be checked. For example, it is not clear why ApMV (correct) has been changed to AMV (incorrect). 

Please sort properly numbers (17 & 18) in the table S2.

Table S2 misses citrus concave gum-associated virus, which was shown to infect apple trees:

Wright, A. A., Szostek, S. A., Beaver-Kanuya, E., & Harper, S. J. (2018). Diversity of three bunya-like viruses infecting apple. Archives of Virology, 163(12), 1–5. http://doi.org/10.1007/s00705-018-3999-z

line 332Eightteen -> Eighteen

line 155genus Ilarvirus',' family Bromoviridae) 

line 224to 'the' genus Ilarvirus

Author Response

Response to Reviewer 1

Nearly all suggestions were addressed. However, the tables still miss a number of recently reported viruses. For example:

Caglayan, K., Roumi, V., Gazel, M., Elci, E., Acioğlu, M., Mavrič Pleško, I., et al. (2019). Identification and Characterization of a Novel Robigovirus Species from Sweet Cherry in Turkey. Pathogens (Basel, Switzerland), 8(2). http://doi.org/10.3390/pathogens8020057

Cao, M., Li, P., Zhang, S., Yang, F., Zhou, Y., Wang, X., et al. (2018). Molecular characterization of a novel citrivirus from citrus using next-generation sequencing. Archives of Virology, 163(12), 3479–3482. http://doi.org/10.1007/s00705-018-4039-8

Marais, A., Faure, C., Theil, S., & Candresse, T. (2018). Molecular characterization of a novel species of capillovirus from Japanese apricot (Prunus mume). Viruses, 10(4). http://doi.org/10.3390/v10040144

Milusheva, S., Phelan, J., Piperkova, N., Nikolova, V., Gozmanova, M., & James, D. (2018). Molecular analysis of the complete genome of an unusual virus detected in sweet cherry (Prunus avium) in Bulgaria. European Journal of Plant Pathology, 153(1), 197–207. http://doi.org/10.1007/s10658-018-1555-z

Koloniuk, I., Sarkisova, T., Petrzik, K., Lenz, O., Přibylová, J., Fránová, J., et al. (2018). Variability Studies of Two Prunus-Infecting Fabaviruses with the Aid of High-Throughput Sequencing. Viruses, 10(4), 204–13. http://doi.org/10.3390/v10040204

Rwahnih, Al, M., Alabi, O. J., Westrick, N. M., & Golino, D. (2018). Prunus geminivirus A: a novel Grablovirus infecting Prunus spp. Plant Disease, 102(7), 1246–1253. http://doi.org/10.1094/PDIS-09-17-1486-RE

Lenz, O., Přibylová, J., Fránová, J., Koloniuk, I., & Špak, J. (2017). Identification and characterization of a new member of the genus Luteovirus from cherry. Archives of Virology, 162(2), 587–590. http://doi.org/10.1007/s00705-016-3125-z

Villamor, D. E. V., Pillai, S. S., & Eastwell, K. C. (2017). High throughput sequencing reveals a novel fabavirus infecting sweet cherry. Archives of Virology, 162(3), 811–816. http://doi.org/10.1007/s00705-016-3141-z

Wu, L. P., Liu, H. W., Bateman, M., Liu, Z., & Li, R. (2017). Molecular characterization of a novel luteovirus from peach identified by high-throughput sequencing. Archives of Virology, 91(9), 1–3. http://doi.org/10.1007/s00705-017-3388-z

We thank the reviewer for drawing our attention to these recent references. These viruses are now included in Tables S1 and S3. The numbers / percentages of viruses have been rectified in the text as appropriate.

 Minor comments:

Abbreviations of viral species should be checked. For example, it is not clear why ApMV (correct) has been changed to AMV (incorrect). 

We thank the reviewer for spotting this error; the abbreviation has been corrected in the revised manuscript.

 Please sort properly numbers (17 & 18) in the table S2.

Table S2 misses citrus concave gum-associated virus, which was shown to infect apple trees:

Wright, A. A., Szostek, S. A., Beaver-Kanuya, E., & Harper, S. J. (2018). Diversity of three bunya-like viruses infecting apple. Archives of Virology, 163(12), 1–5. http://doi.org/10.1007/s00705-018-3999-z

Table S2 has been rectified as requested.

 line 332Eightteen -> Eighteen

line 155genus Ilarvirus',' family Bromoviridae) 

line 224to 'the' genus Ilarvirus

The above typographical errors have been rectified in the revised manuscript.